# An Information Retrieval Approach to Building Datasets for Hate Speech Detection

**Md Mustafizur Rahman**
School of Information
The University of Texas at Austin
nahid@utexas.edu

**Dinesh Balakrishnan**
Department of Computer Science
The University of Texas at Austin
dinesh.k.balakrishnan@utexas.edu

**Dhiraj Murthy**
School of Journalism and Media
The University of Texas at Austin
Dhiraj.Murthy@austin.utexas.edu

**Mucahid Kutlu**
Department of Computer Engineering
TOBB University of Economics and Technology
m.kutlu@etu.edu.tr

**Matthew Lease**
School of Information
The University of Texas at Austin
ml@utexas.edu

## Abstract

Building a benchmark dataset for hate speech detection presents various challenges. Firstly, because hate speech is relatively rare, random sampling of tweets to annotate is very inefficient in finding hate speech. To address this, prior datasets often include only tweets matching known "hate words". However, restricting data to a predefined vocabulary may exclude portions of the real-world phenomenon we seek to model. A second challenge is that definitions of hate speech tend to be highly varying and subjective. Annotators having diverse prior notions of hate speech may not only disagree with one another but also struggle to conform to specified labeling guidelines. Our key insight is that the rarity and subjectivity of hate speech are akin to that of relevance in information retrieval (IR). This connection suggests that well-established methodologies for creating IR test collections can be usefully applied to create better benchmark datasets for hate speech. To intelligently and efficiently select which tweets to annotate, we apply standard IR techniques of *pooling* and *active learning*. To improve both consistency and value of annotations, we apply *task decomposition* and *annotator rationale* techniques. We share a new benchmark dataset for hate speech detection on Twitter that provides broader coverage of hate than prior datasets. We also show a dramatic drop in accuracy of existing detection models when tested on these broader forms of hate. Annotator rationales we collect not only justify labeling decisions but also enable future work opportunities for dual-supervision and/or explanation generation in modeling. Further details of our approach can be found in the supplementary materials [52].

**Content Warning**: We discuss hate speech and provide examples that might be disturbing to read.

## 1   Introduction

Online hate speech constitutes a vast and growing problem in social media [29, 40, 31, 21, 62, 13]. For example, Halevy et al. [29] note that the wide variety of content violations and problem scale on

Facebook defies manual detection, including the rate of spread and harm such content may cause in the world. Automated detection methods can be used to block content, select and prioritize content for human review, and/or restrict circulation until human review occurs. This need for automated detection has naturally given rise to the creation of labeled datasets for hate speech [51, 16].

Datasets play a pivotal role in machine learning, translating real-world phenomena into a surrogate research environments within which we formulate computational tasks and perform modeling. Training data defines the totality of model supervision, while testing data defines the yardstick by which we measure empirical success and field progress. Benchmark datasets thus serve to catalyze research and define the world within which our models operate. However, research to improve models is often prioritized over research to improve the data environments in which models operate, even though dataset flaws and limitations can lead to significant practical problems or harm [71, 35, 41, 58, 47]. Gröndahl et al. [26] argue that for hate speech, the nature and composition of the datasets are more important than the models used due to extreme variation in annotating hate speech, including definition, categories, annotation guidelines, types of annotators, and aggregation of annotations.

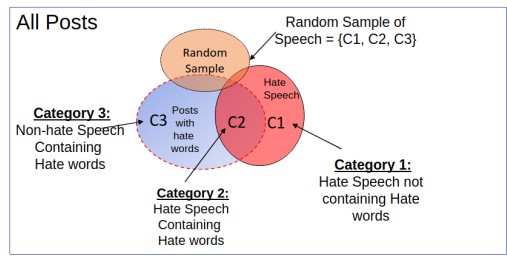

Figure 1: Hate speech coverage resulting from different choices of which social media posts to annotate. Given some list of "hate words" by which to filter posts, some matching posts are indeed hateful (region C2, *true positives*) while other matching posts are benign (region C3, *false positives*). Region C1 indicates *false negatives*: hate speech missed by the filter and mistakenly excluded. Random sampling correctly overlaps C1+C2 but is highly inefficient in coverage.

Table 1: Example tweets from across the Venn diagram regions shown in Figure 1.

| Figure 1 Region | Documents |
| --- | --- |
| C1 | "When you hold all 3 branches and still can't get anything done...you must be a republican." |
| C1 | "You can yell at these Libs all day. They don't listen. But if you ridicule them nationally they can't take it" |
| C2 | "Real Rednecks don't use the internet you pussy" |
| C2 | "guarantee slavery wasnt even tht bad...you know niggas over exxagerate dramatize everything" |
| C3 | "Don't call me black no more. That word is just a color, it ain't facts no more" |
| C3 | "AGENT Are you crazy? You'll never make it in the white trash rap/rock genre with a name like that SKIDMORE ROCKEFELLER" |

Fortunately, many valuable datasets already exist for detecting hate speech [16, 51, 74, 72, 13, 22, 14, 24]. However, each dataset can be seen to embody an underlying design tradeoff (often implicit) in how to balance cost vs. coverage of the phenomenon of hate speech, in all of the many forms of expression in which it manifests. At one extreme, random sampling ensures representative coverage but is highly inefficient (e.g., less than 3% of Twitter posts are hateful [22]). At the other extreme, one can annotate only those tweets matching a pre-defined vocabulary of "hate words" [30, 48] whose presence is strongly correlated with hateful utterances [74, 73, 13, 24, 25]. By restricting a dataset to only those tweets matching a pre-defined vocabulary, a higher percentage of hateful content can be found. However, this sacrifices representative coverage for cost-savings, yielding a biased dataset whose distribution diverges from the real world we seek to model and to apply these models to in practice [32]. If we only look for expressions of hate matching known word lists, the resulting dataset will completely miss any expressions of hate beyond this prescribed vocabulary. This is akin to traditional expert systems that relied entirely on hand-crafted, deterministic rules for classification and failed to generalize beyond their narrow rule sets. Such a resulting benchmark would provide only partial representation for the real-world phenomenon of interest.

**Figure 1** presents a Venn diagram illustrating this. **Table 1** examples further highlight the weakness of only annotating posts matching known "hate words" [13, 74, 25, 24], covering only regions C2-C3. The prevalence of hate found in such datasets is also limited by the word lists used [66].

We can view the traditional practice above as following a two-stage pipeline. Firstly, a simple Boolean search is performed to retrieve all tweets matching a manually-curated vocabulary of known hate words. Secondly, because the above retrieval set is often quite large, random down-sampling is applied to reduce the set to a smaller, more affordable scale for annotation. In information retrieval (IR), a similar pipeline of human annotation following Boolean search has been traditionally used in legal *e-discovery* and medical *systematic review* tasks [36], though the Boolean filter in those domains casts a very wide net to maximize recall, whereas hate word lists tend to emphasize precision to ensure a high percentage (and efficient annotation) of hateful content. However, just as probabilistic models have largely superseded traditional deterministic, rule-based expert systems, we expect that a probabilistic retrieval model can more intelligently and efficiently select which content to annotate.

A second key challenge in constructing hate speech datasets is that what constitutes hate speech is quite subjective, with many competing definitions across legal, regional, platform, and personal contexts [13, 74, 21]. Consequently, annotators having diverse prior notions of hate speech often disagree with one another (especially when labeling guidelines allow more subjective freedom), and may also struggle to conform to stricter labeling guidelines. The use of inexpert crowd annotators can further exacerbate concerns regarding label reliability [63, 47].

A key insight of our study is that the rarity and subjectivity of hate speech are akin to that of relevance in information retrieval (IR) [59]. This suggests that established methods for creating IR *test collections* might also be applied to create better hate speech benchmark datasets. To intelligently and efficiently select which content to annotate for hate speech, we apply two known IR techniques for building test collections: *pooling* [59] and *active learning* (AL) [11, 53]. Both approaches begin with a very large random sample of social media posts to search (i.e., the *document collection*). With pooling, we use existing hate speech datasets and models to train a diverse ensemble of predictive models. We then prioritize posts for annotation by their predicted probability of being hateful, restricting annotation to the resulting *pool* of highly-ranked posts. For nearly 30 years, NIST TREC has applied such pooling techniques with a diverse set of ranking models in order to optimize the coverage vs. cost tradeoff in building test collections for IR, yielding benchmark datasets for fair and robust evaluation of IR systems. AL, on the other hand, requires only an initial set of *seed* posts from which a classifier is progressively trained to intelligently select which posts should be labeled next by human annotators. The tight annotation-training feedback loop provides greater efficiency in annotation, and unlike pooling, it does not require (nor is biased by) existing hate speech datasets.[1] We re-iterate that pooling and AL are established methods in IR; our translational contribution is showing that these techniques can also be usefully applied to efficiently build hate speech datasets without vocabulary filters.

To address subjectivity in annotation, we apply two other techniques from the IR and crowdsourcing literatures. Firstly, applying *task decomposition* [79, 46], we decompose the subjective and complex judging task into a structured set of smaller tasks. Secondly, we design the annotation task to let us measure annotator *self-consistency* [79]. This provides a means of establishing label validity on the basis of an annotator being internally consistent, which is important for subjective labeling tasks in which we expect *inter-annotator* disagreement across annotators. Finally, we require that annotators provide constrained *rationales* justifying their labeling decisions [43, 33], which prior work has found to improve label accuracy, verifiability, and utility.

Using pooling, we create a new benchmark dataset[2] for hate speech detection on Twitter, consisting of 9,667 tweets in total: 4,668 labeled as part of iterative pilot experiments refining quality, and our final high quality set of 4,999 used in our experiments. Each tweet is annotated by three workers on Amazon Mechanical Turk, at a total dataset annotation cost of approximately $5K USD. We assess this dataset with respect to the *prevalence* (i.e., annotation efficiency) and *relative coverage* (breadth) of hate found, as well as evaluation of detection models on these broader forms of hate.

Firstly, we show in Section 3 that pooling yields 14.6% *relative coverage*, far better than the best prior work [22]'s combination of random sampling and keyword filtering (10.4%). Secondly, regarding efficiency in selecting what content to annotate, 14.1% of our annotated content is found to be hateful, a *prevalence* that exceeds a number of prior datasets [13, 21, 25] while simultaneously also providing the aforementioned greater coverage. Note that prevalence can be inflated by only annotating a

---

[1]Strictly speaking, initial seed documents used with AL may also include bias that influences training.

[2]https://github.com/mdmustafizurrahman/An-Information-Retrieval-Approach-to-Building-Datasets-for-Hate-Speech-Detection

highly-restricted vocabulary or set of user posts; e.g., while Waseem and Hovy [74] achieve 31% prevalence, their dataset is highly skewed, with a single user generating 96% of all racist tweets, while another user produces 44% of all sexist tweets [2]. Just as precision and recall are balanced in classification, we wish to balance prevalence (efficiency) and coverage (fidelity) in creating a benchmark dataset that is faithful to the phenomenon while being affordable to create. Finally, we benchmark several recent hate speech detection models [17, 1, 5] and find that the performance of these models drops drastically when tested on these broader forms of hate in our dataset (Table 3).

To further improve annotation efficiency, and to further reduce dependence on prior datasets (and potential bias from them), we also report retrospective experiments that assess AL on our dataset created via pooling. We compare several known AL strategies and feature representations, showing that AL can effectively find around 80% of the hateful content at 50% of the cost of pooling.

As a word of caution, while machine learning models are usually intended to serve the social good (e.g., to reduce online hate speech), it is now well known that machine learning models can also aggravate inequality, perpetuate discrimination, and inflict harm [45, 19]. As noted earlier, one source of such harms are dataset flaws and limitations that are unknown, ignored, or unmitigated [71, 35, 41, 58, 47, 9]. For this reason, we emphasize that hate speech detection researchers and practitioners should attend carefully to these issues when deploying any automated hate speech detection system, including models trained on our dataset. We also want to draw attention to an important issue largely unremarked in prior work on hate speech dataset research: the potential health risks to human annotators from sustained exposure to such disturbing content [67].

Please see our supplementary material [52] for additional discussion of related work, our approach, resulting dataset properties, risks to annotators, and other limitations and risks of our study.

## 2  Methods

To intelligently select which tweets to annotate in order to optimize the prevalence vs. coverage tradeoff in our annotated hate speech dataset, we apply two approaches from constructing test collections in information retrieval (IR): pooling and active learning. Unlike most prior work, we do not use any keywords to filter the set of tweets. Instead, we first collect a large, random (and unbiased) sample of tweets via Twitter's API in which to search for hateful content. Following IR terminology, we refer to each tweet as a *document* and the corpus as a *document collection*.

### 2.1  Selecting Tweets via Pooling

Pooling [59] applies a set of trained machine learning models to predict the hatefulness of documents; the documents that are predicted most likely to be hateful are then selected for human annotation. However, we found that annotating only the top-ranked documents found less diversity in forms of hate than found more broadly online (another example of the *prevalence* vs. *coverage* tradeoff). To further promote diversity, we instead perform a simplified form of stratified sampling. Specifically, we define a minimum threshold $t$, select all tweets with whose predicted hatefulness exceeds $t$ for any model in our ensemble, and then randomly downsample this set to $b$ tweets, given annotation budget $b$. Intuitively, lower values of $t$ inject greater randomness into selection, promiting diversity.

Our pooling algorithm is detailed in **Algorithm 1**. Given document collection $X$, we select a subset of $b$ documents to annotate for our hate speech dataset $R$. Our algorithm requires three inputs: i) a set of prior hate speech datasets $D$; ii) a set of classifiers $C$; and iii) the aforementioned threshold $t$. Given a prior hate speech dataset $D_i$ and a classifier $C_j$, we induce a machine learning model $\hat{C}_j^i$, by training $C_j$ on $D_i$ (Line 4). Then we employ this $\hat{C}_j^i$ to predict a hatefulness score of every document $x$ in the collection $X$ (Line 5). If the predicted score of document $x$ is greater than the provided threshold $t$, we add it to the set $S$ (Lines 6 - 8). This process iterates for all datasets and classifiers in the set $D$ and $C$ (Lines 2 - 8), respectively. Finally, we randomly select $b$ unique documents from $S$ (Line 9), as discussed earlier. to construct the hate speech dataset $R$ (Line 10).

### 2.2  Selecting Tweets via Active Learning (AL)

The underlying assumption of the pooling-based approach is that there exist prior hate speech datasets on which text classification models can be trained to kickstart the pooling technique. However,

**Algorithm 1:** Pooling Approach for Selecting Documents to Annotate

---

**Input** : Document collection $X$ ● Set of prior hate speech datasets D
● Set of classifiers C ● total budget $b$ ● selection threshold $t$
**Output :** Set of documents to be annotated hate speech dataset $R$

**1** $S \leftarrow \emptyset$
**2** **for** $i \leftarrow 1$ **to** $|D|$ **do**
**3**     **for** $j \leftarrow 1$ **to** $|C|$ **do**
**4**         $\hat{C}_j^i \leftarrow$ train_model$(C_j, D_i)$             ▷ train classifier $C_j$ using data $D_i$
**5**         $\forall x \in X$ predict hatefulness of document $x$ using $\hat{C}_j^i$
**6**         **for** $k \leftarrow 1$ **to** $|X|$ **do**
**7**             **if** the predicted hatefulness score of $x^k \geq t$
**8**             **then** $S \leftarrow S \cup x^k$
**9** $R \leftarrow$ Randomly sample $b$ unique documents from $S$        ▷ Set of selected documents

---

this assumption does not always hold [49], especially with less-studied languages (e.g., Amharic, Armenian, etc.). Instead, AL [64, 11, 53] can be applied using the following steps.

**I. Training a Machine Learning Model**. To initiate the active learning process, we need a machine learning model. For AL, we adopt logistic regression as a simple model and two alternative feature representations: i) simple TF-IDF [57]; and ii) BERT embedding from Twitter-roBERTa-base [54].

**II. Defining Document Selection Criteria**. Given a trained model, we can then utilize the predicted posterior probability $p(y^i|x^i)$ of document $x^i$ being hateful to decide whether or not to annotate that document. While uncertainty sampling is most commonly used with AL to select instances to annotate that are close to the decision hyperplane, prior work in AL for IR [11] has shown empirically the benefit of prioritizing selection of examples predicted to come from the rare class when the data is highly skewed. This is because it is important to expose the learner to as many examples of the rare class as possible. In addition, because the model will often be wrong, misses (and near-misses) will still provide ample exposure to ambiguous examples across classes. Because hate speech is quite rare, [22], we thus select those documents for annotation which are most likely to be hateful. Thus we employ *Continuous Active Learning* (CAL) [11, 53], which utilizes $p(y^i|x^i)$ via **Eqn. 1**.

$$x^\star = \operatorname*{argmax}_i \; p(hateful|x^i) \tag{1}$$

We also implement traditional uncertainty sampling [39], referred to in prior IR work as *Simple Active Learning* (SAL) [11], which selects a document for human annotation when the classifier is most uncertain about whether or not a document is hateful. Typically an entropy-based uncertainty function [64] is employed for SAL:

$$Uncertainty(x) = -\sum_{y \in Y} P(y|x) \log P(y|x) \tag{2}$$

where $y$ is either hateful or non-hateful. With this binary classification, SAL selects:

$$x^\star = \operatorname*{argmin}_i \; |p(hateful|x^i) - 0.5| \tag{3}$$

**III. Seed Documents**. Some *seed documents* are necessary to induce the initial machine learning model. Prior work [53] has shown that a few seed documents can suffice to kickstart the AL process. Seed documents can be collected from annotators in various ways: they might write hateful documents (e.g., tweets, posts) or search social media platforms for hateful documents.

The active learning-based process for developing a hate speech dataset is described in **Algorithm 2**. Given seed documents, an initial machine learning model is trained using those seed documents (Lines 1 - 4). The iterative process of AL is conducted in the loop (Lines 5 - 11), where a trained model selects documents for human annotation using the document selection criteria as discussed earlier (Lines 7 - 9). The model is re-trained using both existing documents and those newly annotated documents (Line 10). This continues until the budget is exhausted.

**Algorithm 2:** Active Learning Approach to Selecting Documents for Annotation

---

**Input** : Document collection $X$ ● batch size $u$ ● total budget $b$
**Output :** Hate Speech Dataset $R$

1 Select seed documents $S$
2 $R \leftarrow \{\langle x^i, y^i \rangle \mid x^i \in S\}$                      ▷ Collect initial judgments
3 Learn the initial machine learning model $c$ using $R$
4 $b \leftarrow b - |S|$                              ▷ Update remaining budget
5 **while** *True* **do**
6     **if** $b < u$ **then return**                    ▷ Budget exhausted
7     $\forall x \in X$ predict hatefulness of document $x$ using $c$
8     Select $u$ documents $S \in X$ to judge next
9     $R \leftarrow R \cup \{\langle x^i, y^i \rangle \mid x^i \in S\}$          ▷ Collect judgments
10    Re-train the machine learning model $c$ using expanded $R$
11    $b \leftarrow b - u$                        ▷ Update remaining budget

---

## 3 The Dataset

We create our hate speech dataset via pooling (Section 2.1). We use five models to detect hate speech: logistic regression, naive bayes, an LSTM-based model [5], a BiLSTM-based model [1], and a model based on BERT [17]. For both LSTM and BiLSTM models, we use the improved versions of these models reported by Arango et al. [2]. Models are trained across five prior hate speech datasets: Waseem and Hovy [74], Davidson et al. [13], Grimminger and Klinger [25], Golbeck et al. [24], Basile et al. [6]. Models are trained for binary classification, with datasets binarized accordingly.

**User statistics.** Arango et al. [2] note that 65% of hate speech annotated in Waseem and Hovy [74]'s dataset comes from two Twitter users, suggesting coverage is not representative of online hate. In contrast, our dataset is constructed based on a random sample of tweets from the Twitter API, then selected for annotation via pooling. Out of 9,534 unique users included in our dataset, only 7 produce more than 2 tweets, with at most 15 tweets by one user and 28 total tweets across the other 6.

**Annotation Process**. We collect the annotations in two phases: a set of iterative pilot experiments (4,668 labeled tweets), followed by consistent use of our final annotation design (4,999 labeled tweets). The pilot data revealed two key issues: i) many pornography-related tweets were annotated as hateful, and ii) sometimes annotators did not highlight both the language and targets of hate speech. We thus updated the annotation guidelines accordingly. Although we release the annotated labels collected from both of these phases, we only analyze and report results using the 4,999 annotations collected from the final annotation design.

Since hate speech is highly subjective and complex, prior work [60, 4] has argued that simply asking the annotators to perform binary classification of hate vs. non-hate of an online post is unlikely to be reliable. Thus, a more complex, hierarchical annotation scheme may be needed. For example, Sanguinetti et al. [60] break down the task of annotating hate speech into two sub-tasks. Based on this, we design an annotation scheme that is hierarchical in nature and decomposes the hate annotation task into a set of simpler sub-tasks corresponding to the definitional criteria that must be met for a post to constitute hate speech. Our annotation process is highly structured, including use of term definitions, annotation sub-tasks, types of hate (derogatory language or inciting violence), demographic categories, required rationales for labeling decisions), and a self-consistency test.

**Dataset Properties**. If an annotator identifies (implicit or explicit) language that is derogatory toward or incites violence against a demographic group, and identifies an (implicit or explicit) target group, we can infer from these annotations that a tweet is hateful. In addition, we also ask the annotator (step 6) directly to judge whether the tweet is hateful. This allows us to perform a valuable *self-consistency* check [79] on the annotator, which is especially valuable for subjective tasks in which inter-annotator agreement is naturally lower. We observe that $94.5\%$ of the time, annotators provide final judgments that are self-consistent with their sub-task annotations of 1) hateful language and 2) demographic targets. For the other $5.5\%$ of tweets (274), half of these are labeled as hateful but the annotator does not select either corresponding targets or actions of hate or both, and some annotators still mark pornographic content as hateful. We discard these 274 tweets, leaving 4,725 in our final dataset.

Table 2: Comparison of Datasets. Pooling is seen to achieve the best balance of prevalence vs. relative coverage compared to the keyword-based search (KS) and random sampling (RS) methods.

| Dataset | Size | Method | Prevalence | Rel. Cov. |
|---|---|---|---|---|
| G&K [25] | 2,999 | KS | 11.7% | 0.0% |
| W&H [74] | 16,914 | KS | **31%** | 0.0% |
| G. et al. [24] | 19,838 | KS | 15% | 0.0% |
| D. et al. [13] | 24,783 | KS | 5.77% | 0.0% |
| F. et al. [22] | 91,951 | KS and RS | 4.96% | 10.40% |
| **Our approach** | 4,999 | Pooling | 14.12% | **14.60%** |

Comparable to prior work [50, 55, 34, 15], we observe inter-annotator agreement of Fleiss $\kappa = 0.16$ (raw agreement of 72%) using hate or non-hate binary label of this dataset. Given task subjectivity, this is why self-consistency checks are so important for ensuring data quality. It is also noteworthy that the $\kappa$ score for inter-annotator agreement is not comparable across studies because $\kappa$ depends on the underlying class distribution of the annotators [20]. In fact, $\kappa$ can be very low even though there is a high level of observed agreement [70]. There is in fact debate in the statistical community about how to calculate the expected agreement score [20] while calculating $\kappa$. Gwet [28] advocate for a more robust inter-annotator agreement score and propose $AC_1$, with interpretation similar to the $\kappa$ statistic. Gunther et al. [27] report both Cohen $\kappa$ and Gwet's $AC_1$ in their hate speech dataset and find that the skewed distribution of prevalence of hate speech severely affects $\kappa$ but not Gwet's $AC_1$. We observe $AC_1 = 0.58$ agreement in binary hate vs. non-hate labels. Regarding inter-annotator agreement statistics for rationales supporting labeling decisions, please see our supplementary material [52].

**Relative Coverage**. Since we do not use any "hate words" to filter tweets, our hate speech dataset includes hateful posts from both C1 and C2 categories (Figure 1). To quantify the coverage of hate in a dataset, we quantify the percentage of annotated hateful posts that do not contain known hate words as *Relative Coverage*: $100 \times (N_T - N_{TH})/N_{TH}$, where $N_T$ is the total number of hateful posts in the dataset, and $N_{TH}$ is the total number of hateful posts containing known hate words. For the purpose of analysis only, we use Founta et al. [22]'s hate word list[3]. Any hate speech dataset restricted to tweets having these hate words would have $N_T = N_{TH}$, yielding a relative coverage of 0%, as shown in **Table 2**. Because Founta et al. [22][4] use tweets from both the keyword-based search and random sampling, the relative coverage of their dataset is 10.40%, vs. our 14.60%.

**Prevalence**. Table 2 shows that the prevalence of our hate speech dataset is 14.12%, higher than the datasets created by Founta et al. [22] where the authors cover a broad range of hate speech by combining random sampling with the keyword-based search. In addition to that, the prevalence of our hate speech dataset exceeds that for Davidson et al. [13] and is comparable to the datasets created by Grimminger and Klinger [25] and Golbeck et al. [24], where the authors employ keyword-based search. Again, prevalence can be inflated by only labeling tweets with known strong hate words.

## 4 Model Benchmarking

We benchmark three recent hate speech detection models: LSTM-based [5], BiLSTM-based, [1], and a model based on BERT [17]. For both LSTM and BiLSTM models, we do not use the original versions of these models, but rather the the corrected versions reported by Arango et al. [2].

**Train and Test Sets**. To compare the models, we perform an 80/20 train/test split of our dataset. To maintain class ratios in this split, we apply stratified sampling. For analysis, we also partition test results by presence/absence of known "hate words", using Founta et al. [22]'s hate word list. Train and test set splits contain 3,779 and 946 tweets, respectively. In the test set, the number of normal tweets without hate words and with hate words is 294 and 518 respectively, whereas the number of hateful tweets without hate words and with hate words is 15 and 119, respectively.

**Results and Discussion**. Performance of the models on the test sets in terms of precision (P), recall (R) and $F_1$ is shown in **Table 3**. Following Arango et al. [2], we also report these three performance metrics for both hateful and non-hateful class (Table 3, Column - Class). We discuss the performance of the models considering the two types of test sets as described above.

---

[3]The union of `https://www.hatebase.org` and `https://www.noswearing.com/dictionary`.

[4]Founta et al. [22] report 80K tweets but their online dataset contains 100K. They confirm (personal communication) collecting another 20K after publication. The 92K we report reflects removal of 8K duplicates.

Table 3: Hate classification accuracy of models (bottom 3 rows) provides further support for prior work [2, 3]'s assertion that "hate speech detection is not as easy as we think."

| Method | Class | Without Hate Words | | | With Hate Words | | |
|---|---|---|---|---|---|---|---|
| | | **P** | **R** | **F1** | **P** | **R** | **F1** |
| BiLSTM [1] | Non-Hate | 95.34 | 97.61 | 96.47 | 84.98 | 86.29 | 85.63 |
| LSTM [5] | Non-Hate | 95.60 | 96.25 | 95.93 | 84.95 | 89.38 | 87.11 |
| BERT [17] | Non-Hate | 96.25 | 96.25 | 96.25 | 88.84 | 83.01 | 85.82 |
| BiLSTM [1] | Hate | 12.50 | 6.666 | 8.695 | 36.03 | 33.61 | 34.78 |
| LSTM [5] | Hate | 15.38 | 13.33 | 14.28 | 40.21 | 31.09 | 35.07 |
| BERT [17] | Hate | 26.66 | 26.66 | 26.66 | 42.48 | 54.62 | 47.79 |

**Case I. Without Hate Words**. Intuition is that correctly classifying a document as hateful when canonical hate words are absent is comparatively difficult for the models. This is because models typically learn training weights on the predictive features, and for hate classification, those hate words are the most vital predictive features which are absent in this setting. By observing $F_1$, we can find that for the hateful class, all these models, including BERT, provide below-average performance ($F_1 \leq 26.66\%$ ). In other words, the number of false negatives is very high in this category. On the other hand, when there are no hate words, these same models provide very high performance on the non-hateful class, with $F_1 \approx 96\%$.

**Case II. With Hate Words**. In this case, hate words exist in documents, and models are more effective than the previous case in their predictive performance (Table 3, Columns 6, 7, and 8) on the hateful class. For example, $F_1$ of BERT on the hateful class has improved from 26.66% (Column 3) to 47.79% (Column 8). This further shows the relative importance of hate words as predictive features for these models. On the other hand, for classifying documents as non-hateful when there are hate words in documents, the performance of models are comparatively low (average $F_1$ across models is $\approx 86\%$) vs. when hate words are absent (average $F_1$ across models is $\approx 96\%$).

**Table 3** shows clearly that models for hate speech detection struggle to correctly classify documents as hateful when "hate words" are absent. While prior work by Arango et al. [2] already confirms that there is an issue of overestimation of the performance of current state-of-the-art approaches for hate speech detection, our experimental analysis suggests that models need further help to cover both categories of hate speech (C1 and C2 from Figure 1).

## 5 Active Learning vs. Pooling

While pooling alone is used to select tweets inclusion and annotation in our dataset, we also report a retrospective evaluation of pooling vs. active learning (AL). In this retrospective evaluation, AL is restricted to the set of documents selected by pooling, rather than the original full, random sample of Twitter from which the pooling dataset is derived. Because of this constraint, the retrospective AL results we report here are likely lower than what might be achieved when AL run instead on the full dataset, given the larger set of documents that would be available to choose from for annotation.

**Experimental Setup**. Each iteration of AL selects one document to be judged next (i.e., batch size $u$ = 1). The total allotted budget $b$ is set to the size of the hate speech dataset constructed by pooling ($b = 4,725$). For the document selection, we report SAL and CAL. As a baseline, we also report a random document selection strategy akin to how traditional supervised learning is performed on a presumably random sample of annotated data. Following prior work's nomenclature, we refer to this as *simple passive learning* (SPL) [11]. As seed documents, we randomly select 5 hateful and 5 non-hateful documents from our hate speech dataset constructed using pooling.

**Experimental Analysis**. We present the results (**Figure 2**) as plots showing the cost vs. effectiveness of each method being evaluated at different cost points (corresponding to varying evaluation budget sizes). We also report Area Under Curve (AUC) across all cost points, approximated via the Trapezoid rule. To report the effectiveness, Figure 2 presents $F_1$ performance (left plot), and prevalence (right plot) results of the three document selection approaches: SPL, SAL, and CAL, along with two feature representations: TF-IDF and contextual embedding from BERT.

While reporting $F_1$ of our AL classifier (Figure 2, left plot), following prior work [44], we consider both human judgments and machine predictions to compute $F_1$. For example, when we collect 20%

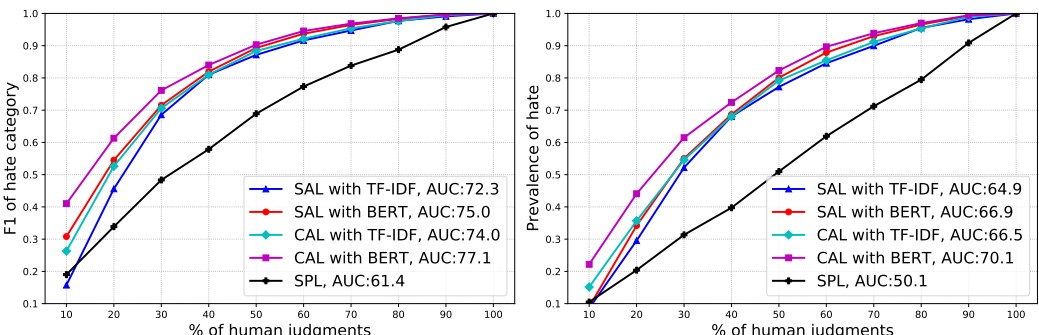

Figure 2: Left plot shows human judging cost (x-axis) vs. F1 classification accuracy (y-axis) for hybrid human-machine judging. Right plot shows human judging cost vs. prevalence of hate speech for human-only judging of documents. The % of human judgments on x-axis is wrt. the number documents in the hate speech data constructed by the pooling-based approach.

human judgments from annotators, the remaining 80% is predicted by the AL classifier (i.e., machine prediction). Then those two sets (human judgments and machine prediction) are combined to report the final $F_1$ score. However, to report the prevalence of hate (right plot), only human judgments are considered because we need the actual labels of documents, not the machine predictions here.

**Active vs. Passive Learning**. By comparing active learning (SAL and CAL) methods against passive learning (SPL) method in terms of $F_1$, we find that SAL and CAL significantly outperform SPL in terms of AUC. The observation holds for both types of feature representations of documents. The only exception we can see that at 10% human judgments, $F_1$ of SPL is slightly better than SAL with TF-IDF. Nevertheless, after that point of 10% human judgments, SAL with TF-IDF consistently outperforms SPL in terms of $F_1$. Similarly, if we consider the prevalence of hate (right plot), active learning exceeds passive learning by a large margin in AUC.

**CAL vs. SAL**. From Figure 2, we can see that CAL consistently provides better performance than SAL both in terms of $F_1$ and prevalence. In fact, when the allotted budget is very low (e.g., budget $\leq$ 20% of human judgments), the performance difference in terms of $F_1$ and prevalence between CAL and SAL considering the underlying feature representation is very high.

**TF-IDF vs. BERT embedding**. It is evident from Figure 2 that the contextual embedding from BERT for document representation provides a significant performance boost over the TF-IDF-based representation. SAL and CAL both achieve better performance when the document is represented using BERT. Furthermore, we can find that for a low-budget situation (e.g., budget $\leq$ 20% of human judgments), CAL with BERT provides the best performance.

**Judging cost vs. Performance** We also investigate whether the AL-based approach can provide better prevalence and classification accuracy at a lower cost than the pooling-based approach. From Figure 2, we can find that the AL-based approach achieves $F_1 \geq 0.9$ (left plot) and finds 80% of hateful documents (right plot) by only annotating 50% of the original documents judged via pooling.

## 6 Conclusion & Future Work

The success of an automated hate speech detection model largely depends on the quality of the underlying dataset upon which that model is trained on. However, keyword-based filtering approaches [13, 74, 25] do not provide broad coverage of hate, and random sampling-based approaches [32, 14] suffer from low prevalence of hate. We propose an approach that adapts *pooling* from IR [59] by training multiple classifiers on prior hate speech datasets and combining that with random sampling to improve both the prevalence and the coverage of hate speech in the final constructed dataset.

Using the pooling technique, we share a new benchmark dataset for hate speech detection on Twitter. Results show that the hate speech dataset developed by applying the proposed pooling-based method achieves better relative coverage (14.60%) than the hate speech dataset constructed by Founta et al. [22] that combines random sampling with the keyword-based search (Table 2). Furthermore, the prevalence of hate-related content in our hate speech dataset is comparable to many prior keyword-

based approaches [13, 25, 24]. We also show a dramatic drop in accuracy of existing detection models [17, 5, 1] when tested on these broader forms of hate.

An important limitation of the pooling approach is that it relies on prior hate speech datasets. That said, though the datasets used to train our pooling prediction models may lack diverse tweets, the trained models still learn correlations between hate labels and all vocabulary in the dataset. This enables pooling to identify some hateful tweets (for inclusion in our dataset) that lack known hate words, though this prediction task is clearly challenging (as the bottom left of Table 3 indicates). In general, the purpose of pooling in IR is that diverse models better identify potentially relevant content for human annotation. In the translational use of pooling we present for hate speech, training a cross product of different models across different prior datasets similarly promotes diversity and "wisdom of crowds" in identifying potentially hateful content for inclusion and annotation.

To sidestep this reliance on prior hate speech datasets for pooling, we also present an alternative approach that utilizes active learning [64] to develop the hate speech dataset. Empirical analysis on the hate speech dataset constructed via the pooling-based approach suggests that by only judging 50% of the originally annotated documents, it is possible to find 80% of hateful documents with an $F_1$ accuracy of $\geq 0.9$ (Figure 2) via our AL-based approach.

Like HateXplain [42], our collection of rationales as well as labels creates the potential for explainable modeling [69] and dual-supervision [78]. However, while HateXplain collects rationales only for the overall labeling decision, our collection of rationales for different annotation sub-tasks creates intriguing possibilities for dual-supervision [76] and explanations across different types of evidence contributing to the overall labeling decision [38]. In particular, annotator rationales identify i) derogatory language, ii) language inciting violence, and iii) the targeted demographic group. As future work, we plan to design dual-supervised [76] and/or explainable [18] machine learning models that can incorporate the annotators' rationales collected in our hate speech dataset.

Our definition of hate speech and annotation process assumed that hate speech is composed of two parts: 1) language that is derogatory or inciting violence against 2) a target demographic group. However, an interesting question is whether perceptions of hate differ based on the demographic group in question, e.g., a given derogatory expression toward a political group might be deemed acceptable while the same expression to a racial or ethnic group might be construed as hate speech. HateCheck [56]'s differentiation of general templates of hateful language vs. template instantiations for specific demographic targets could provide a nice framework to further investigate this.

Although our annotation process was highly structured, we ultimately still produced binary labels for hate speech, lacking nuance of finer-grained ordinal scales or categories. It would be interesting to further explore our structured annotation process with such finer-grained scales or categories.

Recent years have brought greater awareness that machine learning datasets (as well as models) can cause harm as well as good [65, 10, 45, 19]. For example, harm could come from deploying a machine learning model without considering, mitigating, and/or documenting [23, 7, 9] limitations of its underlying training data, such as the risk of racial bias in hate speech annotations [61, 75, 12]. Both researchers and practitioners of hate speech detection should be well-informed about such potential limitations and risks of any constructed hate speech dataset (including ours) and exercise care and good judgment while deploying a hate speech detection system trained on such a dataset.

We know of no prior work studying the effect of frequent exposure to hate speech on the well-being of human annotators. However, prior evidence suggests that exposure to online abuse has serious consequences on the mental health of workers [77]. Studies by Boeckmann and Liew [8] and Leets [37] to understand how people experience hate speech found that low self-esteem, symptoms of trauma exposure, etc., are associated with repeated exposure. The Linguistic Data Consortium (LDC) reported its annotators experiencing nightmares and other overwhelming feelings from labeling news articles [68]. We suggest more attention be directed toward the well-being of the annotators [67].

**Acknowledgements**. We thank the many talented Amazon Mechanical Turk workers who contributed to our study and made it possible. This research was supported in part by Wipro (HELIOS), the Knight Foundation, the Micron Foundation, and Good Systems (`https://goodsystems.utexas.edu`), a UT Austin Grand Challenge to develop responsible AI technologies. Our opinions are our own.

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
