# OpenReview forum: "An Information Retrieval Approach to Building Datasets for Hate Speech Detection"
_NeurIPS.cc/2021/Track/Datasets_and_Benchmarks/Round2 — NeurIPS 2021 Datasets and Benchmarks Track (Round 2)_

### Official Review · Reviewer_D4Fi · 2021-09-15
**A helpful new dataset on hate speech that still has room for improvement**

**Rating:** 7
**Confidence:** 4
**Correctness:** Yes.
**Clarity:** The paper is well written.

**Strengths:**

To address some of the limitations of prior research, an alternative and viable technique is utilized. This method is based on the Information Retrieval method, which is an approach that has been neglected in the past. Additionally, this research used task decomposition, which may assist in the annotation of hate speech messages and improve annotator agreement in future investigations.

**Weaknesses:**

Annotation guidelines could be improved as they induce some behavior in the annotators.  First, guidelines request that annotators must identify both the targets and the actions related to hate speech otherwise, their work will be rejected. Nevertheless, not all hostile texts include one or both of these components. Second, also highlighting all words in the text will yield rejection, which may also lead users to not annotate all the desired content as they would wish.

Despite previous studies warning (Arango et al. [3]), this study does not provide statistics on the data authors. It is critical to have an anonymised id for the message's author, so that data from the same user may only be utilized for training or testing purposes. Statistics such as the total number of users and the total number of messages sent by each user should be given - either as the distribution of this value within the sample or as the mean and standard deviation.

It is not clear how both pooling and active learning are combined in order to choose data for annotation.

Limited self-consistency metric: this was measured by when there's target highlight the author should mark the message as containing hate speech. This happened 94.5% of the time. However, there are hate speech messages that don’t contain direct targets, for instance: “Not all human beings deserve same rights. Some are closer to behave like animals.” Additionally, the authors discuss how self-consistency may be utilized when annotator agreement is low. Both, however, relate to distinct constructs.

It is unclear which questions were used to calculate the annotator agreement. We have a binary metrics for hate speech in this job, but we also have annotation at the word level, for which it would be beneficial to understand how agreement is calculated. Additionally, it is unclear how the responses of three annotators result in the annotation of the message for target and action.


**Additional Feedback:**

Few corrections:

“representative coverage for cost-savings, yielding a biased dataset *whose* whose distribution diverges”

“incites *violates* against a demographic group, and identifies that target group”

In more than one case appears:
“For a tweet, if an annotator identifies *langauge* that is derogatory toward or”


**Documentation:**

Yes to both.

**Ethics:**

No ethical concerns with this study.

**Relation To Prior Work:**

It discusses the comparison with previous works.

**Summary And Contributions:**

The article describes the annotation of a new hate speech dataset (9,667 tweets in total) with the objective of addressing the limitations of keyword-based filtering methods for collecting hate speech data. As a solution, methods for information retrieval are employed.
The annotation process starts by searching a random sample of social media postings. The data is subsequently chosen via pooling and active learning. The data is then annotated by untrained Mechanical Turk crowdworkers. To help workers improve their annotation quality, the article applies task decomposition to hate speech and proposes to measure annotator self-consistency. A benchmarking is performed using the new data and models based on LSTM, BiLSTM, and BERT.

---

> ### Author Response · Authors · 2021-09-28
> **Reply to Reviewer 4**
>
> We thank the reviewer for their time and valuable feedback. We are glad that the reviewer found the paper to be well-written and that they appreciated the IR method used to address limitations of prior research, and the use of task decomposition for structured annotation of hate speech.
>
> **“guidelines request that annotators must identify both the targets and the actions related to hate speech otherwise, their work will be rejected...” and later “there are hate speech messages that don’t contain direct targets.”** Workers who labeled a tweet as hateful were required to support their labeling decision by marking either implicit or explicit hate or targets. We did not assume that hate and targets are explicit. In Section, steps 3-4 specify how to label explicit vs. implicit hate and targets. Self-consistency checks only required that either implicit or explicit evidence was provided for labeling a tweet as hateful. We have clarified this in Section 3.
>
> **“highlighting all words in the text will yield rejection”**. We ask workers to highlight words or phrases related to three specific types of content: “INCITING VIOLENCE”, “DEROGATORY LANGUAGE”, and “the INTENDED TARGET”.  In the original pilot task, we observed that workers sometimes highlighted the entire text of a tweet, and that this nearly always corresponded to low quality work and failure to follow instructions. Consequently, we added the specific warning identified by the reviewer. We have added discussion of this in Section D.3.
>
> **“It is critical to have an anonymised id for the message's author, so that data from the same user may only be utilized for training or testing purposes”**. Because hate speech is relatively rare, Waseem and Hovy “identified a small number of prolific users” and collected tweets from them. This led to Arango et al.’s critique that a single user generated 96% of W&H’s racist tweets, with the same user’s tweets appearing in both train and test. In contrast, we began with a uniform random sample of tweets, then selected tweets to annotate by pooling across classifiers. Tweets from the same user are thus quite rare in our dataset. Specifically, out of 9,534 unique authors, 9,430 write 1 tweet, 97 do 2 tweets, and only 7 have more than 2 tweets, with a max of 15 tweets by one author and 43 total tweets across the 7 authors. We have added these statistics to Section 3.
>
> **“It is not clear how both pooling and active learning are combined in order to choose data for annotation.”** Tweets were selected for annotation using pooling alone, and our analysis using active learning was performed retrospectively on our labeled dataset. We have further clarified this in Section 5.
>
> **“the authors discuss how self-consistency may be utilized when annotator agreement is low. Both, however, relate to distinct constructs.”** We are not familiar with any prior work on hate speech annotation using self-consistency checks, which we believe complements traditional use of annotator agreement measures. Conceptually, for objective tasks with a single true answer, we expect reasonable annotator agreement, but on subjective tasks (e.g., favorite ice cream flavor) we do not expect annotators to agree.  While an annotator can be expected to be self-consistent for either task type, self-consistency seems particularly valuable for subjective tasks when annotators are expected to disagree with one another. Hate speech annotation lies in the spectrum between objective vs. subjective tasks. Some objectivity is necessary to yield consistent data for training detection models, but low annotator agreement remains common. This is why we believe self-consistency measures can complement traditional practice here. We have added this discussion to Section B.6.
>
> **“It is unclear which questions were used to calculate the annotator agreement.”** The fleiss kappa annotator agreement statistic we reported was for question 6 in Section D.2: “do you believe the post is hateful? [YES / NO]”. We did not compute annotator agreement at the word-level in annotating rationales, though we have also added the rationale annotations to our github repo, allowing measurement of fleiss kappa at the word level as in [Eraser](https://arxiv.org/abs/1911.03429). How to compute overall annotator agreement across structured annotation tasks with subtasks remains an open research question.
>
> **“it is unclear how the responses of three annotators result in the annotation of the message for target and action.”** To aggregate individual annotator labels to induce consensus, we report majority voting for categorical annotations of binary hate (Section D.3) and target group (Section F). While not performed in our study, word-level annotations could be aggregated using sequence-label aggregation methods from prior work: [Nguyen et al., ACL’17](https://www.ischool.utexas.edu/~ml/papers/nguyen-acl17.pdf) or [Braylan and Lease, Web’20](https://www.ischool.utexas.edu/~ml/papers/braylan_web2020.pdf).

---

> > ### Comment · Reviewer_D4Fi · 2021-09-29
> > **It might be beneficial to make a few more adjustments.**
> >
> > Thank you so much to the writers for taking the time to respond.
> > I believe we are headed in the right path for me to improve my score. However, there are still certain issues that need to be solved, including the following:
> >
> > - “It is critical to have an anonymised id for the message's author, so that data from the same user may only be utilized for training or testing purposes” I acknowledged the new addition. However, considering the mistakes and drawbacks in previous datasets, I think we need to push forward quality standards in hate speech research. You make clear now that  "Specifically, out of 9,534 unique authors, only 7 have more than 2 tweets, with a max of 15 tweets by one author and 43 total tweets across the 7 authors. We have added these statistics to Section 3.". What if all the remaining authors have produced 2 messages each? New researchers using your dataset will not have a way to assure data from the same user is only utilized for training or testing purposes. I think you need to provide anonymized author ids.
> >
> > - “It is not clear how both pooling and active learning are combined to choose data for annotation.” Can you please elaborate further on the explanation you have provided?
> >
> > - “It is unclear which questions were used to calculate the annotator agreement.” The authors replied "We did not compute annotator agreement at the word-level in annotating rationales, though we have also added the rationale annotations to our github repo, allowing measurement of fleiss kappa at the word level as in Eraser." Are you able to include this statistic in your article as well? Because the rationales are such an essential element of your annotation, I believe it would be very beneficial.

---

> > > ### Author Response · Authors · 2021-09-30
> > > **Additional comments for Reviewer 4**
> > >
> > > Thank you for your interest in our work and the quick response!  It's great to have a reviewer very engaged with us in the review process.
> > >
> > > **Distribution of authors and tweets**. Because the dataset contains 9,667 tweets across 9,534 distinct authors, we know that most authors write only a single tweet: only 9,667-9,534=133 tweets in our dataset, or <1.4% total, represent additional tweets. More specifically, 9,430 authors write 1 tweet, 97 authors write 2 tweets, and of the 7 authors writing more than 2 tweets, they collectively post only 43 total tweets, with a max of 15 tweets by a single author. We could add author IDs as requested, but given the numbers here, it was not clear to us how useful this would be to other researchers.
> > >
> > > **Selection of tweets for annotation via pooling.** We apologize for the continued confusion here. We will revise the abstract and introduction for the camera-ready version of our paper (if accepted) to further clarify that Tweets were selected for annotation (i.e., inclusion in our dataset) based entirely on pooling. After creating the dataset via pooling, we then later went back and did a retrospective evaluation of AL that found that AL could effectively have found around 80% of the hateful content at 50% of the cost of pooling. Section 5's "Active Learning vs. Pooling" discusses our AL work. "While pooling alone is used to select tweets for annotation in our dataset, we evaluate AL methods retrospectively vs. pooling. To understand the effectiveness of AL, we set the document collection X of Algorithm 2 to the hate speech dataset constructed by the pooling-based approach. In other words, AL can only request labels for documents which are already in our hate speech dataset constructed via pooling, thus results provide a lower-bound for what might be achieved with unconstrained selection via AL."
> > >
> > > If further clarification on pooling vs. AL would help, please feel free to ask.
> > >
> > > **"We did not compute annotator agreement at the word-level in annotating rationales, though we have also added the rationale annotations to our github repo, allowing measurement of fleiss kappa at the word level as in Eraser." Are you able to include this statistic in your article as well?**.  Yes, we can add this for the camera-ready version of our paper (if accepted).

---

> > > > ### Comment · Reviewer_D4Fi · 2021-09-30
> > > > **Final Reply**
> > > >
> > > > Distribution of authors and tweets -  I understand the percentage you provide looks small (<1.4%). However, it is error that you will introduce in the classifier and leave no way to control it. Additionally, what if the messages from same author belong all to the hate speech class? (the 1.4% percentage will become more relevant as it will contribute for differences between both classes).
> > > >
> > > > Selection of tweets for annotation via pooling - Now that it is clear that the new dataset is produced only by pooling and using classifiers trained with previously annotated datasets, I believe that this method, given the lack of generalization of hate speech classifiers proven in previous studies, increases the likelihood that data in this dataset will be biased and more homogeneous and similar to previous datasets.

---

### Official Review · Reviewer_ncnt · 2021-09-17
**Improving hate speech detection datasets**

**Rating:** 9
**Confidence:** 5
**Correctness:** I am satisfied with the correctness o…
**Clarity:** Yes

**Strengths:**

- The paper is rigorous and it is clear that the authors have a strong grasp of the limitations of current work in this area.
- The authors cite most of the relevant literature.
- This dataset will be useful for researchers working in this area.
- The authors carefully consider the ethical implications of using crowdsourced hate speech annotations.

**Weaknesses:**

- Some of the examples used are questionable. For example, Table 1 contains the following example of "hate speech not containing hate words": “You can yell at these Libs all day. They don’t listen. But if you ridicule them nationally they can’t take it". This is clearly a derogatory comment towards a political opponent (presumably a conservative criticizing liberals), but this seems very different to other examples here. Just criticizing a political opponent using a nickname is not hate speech. This raises some concerns about the validity of the annotated data.
- I was surprised that the authors develop a binary hate speech classifier given that much of the previous work in the area has criticized such approaches. The authors mention this in the Appendix and state it is a limitation and how it caused problems in the present study (pornographic references misclassified as hate speech). I would like to see a better justification of the binary classifier approach and an analysis of how this approach performs when applying it to data with multiple labels (e.g. Founta et al's dataset).
- Like much of the research on online hate speech detection, this paper uses data from Twitter. It is not clear that this benchmark will be useful for models trained on other datasets. The contribution is therefore more limited than stated: this is a benchmark for Twitter hate speech detection models.

**Additional Feedback:**

N/A

**Documentation:**

The authors link to a Github repository containing the annotations. I'm a little concerned that only the tweet IDs are provided and not the original text. This is particularly important in this context since many of the hate speech tweets will be likely to be deleted. I believe it is within the Twitter terms of service to release more information on the tweets.

**Ethics:**

The authors carefully consider relevant ethical issues.

**Relation To Prior Work:**

Yes

**Summary And Contributions:**

This paper proposes a new dataset for hate speech detection. The authors use two different methodologies, pooling and active learning, to derive a dataset consisting of tweets annotated for hate speech. This approach builds upon datasets collected by several other scholars. This dataset is intended to serve as a more reliable benchmark for hate speech detection models.

---

> ### Author Response · Authors · 2021-09-28
> **Reply to Reviewer 3**
>
> We thank the reviewer for their time and valuable feedback. We appreciate that the reviewer found our work rigorous  and ethically grounded. The reviewer further noted that the dataset we produced will be useful for researchers.
>
> **“Like much of the research on online hate speech detection, this paper uses data from Twitter… The contribution is therefore more limited than stated: this is a benchmark for Twitter hate speech detection models.”** Fair point, we have clarified ours is a Twitter dataset in the Abstract, Introduction, Conclusion, and Appendix H.
>
> **" 'You can yell at these Libs all day. They don’t listen. But if you ridicule them nationally they can’t take it'. This is clearly a derogatory comment...  but this seems very different to other examples here. Just criticizing a political opponent using a nickname is not hate speech.”** There is great disagreement in prior work and across annotators in both how hate speech is defined across studies/platforms, as well as how it is perceived by different people/annotators. Let us consider the example here in light of the recent [HateCheck](https://arxiv.org/abs/2012.15606) paper. They create various templates (e.g., “if you ridicule [GROUP] they can’t take it.”) and instantiate these templates with different demographic groups -- e.g., political (libs, conservatives), religious (Christians, Muslims, Jews), gender (e.g., women, men, transgender), skin color (whites/blacks/browns), etc.). As the reviewer here notes, the language in this example is clearly derogatory, yet perhaps whether it is hate speech depends on which target demographic instantiates the template. We might thus investigate how judgments are shaped by language used vs. target group. We have added this discussion to the Conclusion.
>
> **“I would like to see a better justification of the binary classifier approach and an analysis of how this approach performs when applying it to data with multiple labels (e.g. Founta et al's dataset).”** As a clarification, we believe this question is referring to the binary annotation scheme, rather than automatic classification, given the mention of “pornographic references misclassified”, which refers to errors we observed from our human annotators.
>
> Firstly, it is important to distinguish the output categorization (e.g., binary hate vs. non-hate) vs. the sophistication of the process used to arrive at these categories. As a simple example, one could use a process asking annotators to label a 3, 5, or 7 degree scale of hate and then collapse that to binary hate/non-hate categories. This might lead to more consistent binary judgments than collecting binary judgments directly. While distinguishing degrees or types of hate in output categories is clearly valuable, as we note in the appendix, the primary issue we raise and seek to remedy is the overly simplistic annotation process often found in prior work. When one simply asks an annotator “hateful or not” for a given tweet, this is highly problematic.
>
> For example, in B.3 we wrote, “Most of the time, researchers only specify the definition of categories (e.g., hate or offensive) [21, 32] but do not provide any additional clarification about how to interpret each of those categories.” We do not simply ask annotators to provide a binary label, but as discussed in Appendix D, we highly structure the annotation process. In D.2 we wrote, “Based on Sanguinetti et al. [87], we design an annotation scheme that is hierarchical in nature and decomposes the hate annotation task into a set of simpler sub-tasks corresponding to the definitional criteria that must be met for a post to constitute hate speech.” We further perform self-consistency checks to ensure that sub-tasks agree with the final overall label. In addition, we require annotators to highlight specific terms for derogatory language and the target demographic group, as explicit rationales supporting labeling decisions.
>
> Because our annotation process was already highly structured -- including use of definitions, annotation sub-tasks, types of hate (derogatory language or inciting violence), demographic categories, and required rationales for labeling decisions), we did not ask our Turkers to further label degree of hate. We agree that it would be interesting to also have that output, but our focus was on implementing a highly rigorous annotation process that did not overwhelm our Turkers. We have added further language about this Section 3.
>
> Regarding “how this approach performs when applying it to data with multiple labels”, we do agree that this would be very interesting.  However,  this is beyond the remit of our current dataset and study. We now suggest this as future work in our Conclusion.
>
> **“I'm a little concerned that only the tweet IDs are provided and not the original text.”** [Twitter policy](https://developer.twitter.com/en/developer-terms/more-on-restricted-use-cases)  explicitly prohibits this. See also our related reply to Reviewer 2.

---

### Official Review · Reviewer_YDBN · 2021-09-19
**An Information Retrieval Approach to Building Datasets for Hate Speech Detection**

**Rating:** 8
**Confidence:** 3

**Strengths:**

* Hate detection is a challenging, yet very relevant/important topic with positive social implications.
* Existing datasets are created from a predefined sets of words, whereas the proposed dataset, consist of hateful tweets with and without hateful words - making it more diverse and closer to real-world applications.
* The authors present and Active Learning strategy for creating new datasets for hate detections and show that it is very efficient (in terms of annotation costs). This will pave the way for researchers to create larger datasets at reasonable costs.

**Weaknesses:**

* The dataset is rather small. It consists of only 10k tweets, whereas the largest existing dataset has 90k tweets. However, I still think the dataset is relevant as it is of higher quality (to the best of my knowledge)
* As the authors already point out the proposed pooling method is biased by existing datasets. They proposed a stratified sampling to reduce this bias.

**Additional Feedback:**

See above comments.

**Clarity:**

The paper is clear and well written. There are some small typos (I would recommend pasting the text in Google docs and use their spelling control to correct these). Examples:

In the abstract: "  for hate speech To int " => "  for hate speech. To int"
In the intro: " biased dataset whose whose distribution" => " biased dataset whose distribution"

Also, I found the following sentence unclear. Please reformulate:

"Train and test set splits contain 3,779 and 637 / 309 tweets, respectively."




**Correctness:**

To the best of my knowledge, the dataset is constructed in a well considered and correct way. A 20/80 test/train division is performed and sampled with stratified sampling, which ensures that each category of hate tweets are considered in both test/train set. Reasonable baseline methods are chosen.

**Documentation:**

A link to the dataset is provided. Here the tweets are listed with binary labels. However, I don't see the actual tweets? Where are the text corresponding to the ID stored? I think it is important to release other meta-data with the tweets besides the binary labels. I would like that all the annotated data was released including target, group identity etc.

**Ethics:**

Yes, the authors have both disclaimer and discussion on biases and uses.

**Relation To Prior Work:**

Yes, this is quite clear. One thing that could be improved is to train the methods on an existing dataset and see how well the performance is on the proposed dataset. As I understand, in the current experimental setup, you train/test only on the proposed dataset. It is interesting to understand if the performance of the models are improved by a bigger training set (of lower quality) or if the lower quality degenerates the performance when tested on your dataset.

**Summary And Contributions:**

A new dataset for hate speech detection is presented. In contrast to prior works that are collected with a keyword based approach, this dataset is collected with a pooling based approach. This results in a dataset that contains hateful tweets without containing a predefined set of hateful words (in contrast to prior works), making the dataset highly relevant and closer to real-world applications. The authors exemplifies this by showing that SOTA methods perform poorly on this dataset, especially for detecting hateful tweets that does not contain hateful words (F1 < 27 %). Furthermore, the authors present an Active Learning strategy and shows that it could be very useful to create hate speech datasets in the future.

---

> ### Author Response · Authors · 2021-09-28
> **Reply to Reviewer 2**
>
> We thank the reviewer for their time and helpful feedback. It was great to see that the reviewer appreciated having a hate speech dataset closer to real-world applications by covering hateful tweets lacking predefined hateful words (in contrast to prior works). The reviewer also noted the value in the active learning strategy we proposed, given that current methods perform poorly in detecting hateful tweets that do not contain hateful words (F1< 27%).
>
> **“As the authors already point out, the proposed pooling method is biased by existing datasets. They proposed a stratified sampling to reduce this bias.”** Although existing datasets have been largely collected by matching known hate words (which does naturally influence the prediction models trained on these datasets), the trained models still do learn correlations between hate labels and the entire vocabulary in each dataset. This enables pooling to identify some hateful tweets (for inclusion in our dataset) that lack known hate words. Though, this prediction task is clearly challenging (as the bottom left of Table 3 indicates). In general, the purpose of pooling in IR is that a combination of diverse models can better identify potentially relevant content for human annotation. In the translational use of pooling we present for hate speech, training a cross product of different models across different prior datasets similarly promotes diversity and “wisdom of crowds” in identifying potentially hateful content for inclusion and annotation. That said, pooling is still limited by the set of available training datasets, as noted. As the reviewer mentions, however, the stratified sampling helps to further alleviate this.
>
> **“One thing that could be improved is to train the methods on an existing dataset and see how well the performance is on the proposed dataset.”** Yes, we did not compare how in-domain vs. out-of-domain training performs, or transfer learning / domain-adaptation more generally.  We agree that this would be an interesting direction for future work, particularly in regard to mismatch between training and testing data in their coverage of diverse forms of hate speech (i.e., with or without coverage of hateful tweets lacking known hate words).
>
> **“I don't see the actual tweets? Where are the text corresponding to the ID stored?”**. Unfortunately, Twitter prohibits sharing the tweet text directly, though others have posted tweet texts in violation of [Twitter policy](https://developer.twitter.com/en/developer-terms/more-on-restricted-use-cases): “If you need to share Twitter content you obtained via the Twitter APIs with another party, the best way to do so is by sharing Tweet IDs, Direct Message IDs, and/or User IDs, which the end user of the content can then rehydrate (i.e. request the full Tweet, user, or Direct Message content) using the Twitter APIs… We permit limited redistribution of hydrated Twitter content via non-automated means. If you choose to share hydrated Twitter content with another party in this way, you may only share up to 50,000 hydrated public Tweet Objects and/or User Objects per recipient, per day, and should not make this data publicly available (for example, as an attachment to a blog post or in a public Github repository).” We also mention this issue in Appendix Section B.7 regarding limitations of shared Twitter datasets for hate speech research.
>
> **“I would like that all the annotated data was released including target, group identity etc.”** We apologize for missing this in our initial dataset release and have now corrected this.
>
> **"I found the following sentence unclear. Please reformulate: Train and test set splits contain 3,779 and 637 / 309 tweets, respectively."**. This is now revised as requested in Section 4 paragraph 2.

---

### Official Review · Reviewer_jqPw · 2021-09-20
**Interesting methodological considerations but a somewhat disappointing dataset in the end**

**Rating:** 5
**Confidence:** 5
**Correctness:** The claims are globally correct but s…

**Strengths:**

- the limitations of existing hate speech datasets are well explained, in particular the lack of diversity of datasets aggregated through a keyword-based approach
- the paper contains interesting methodological considerations on how to create a more diverse dataset in an efficient way. In particular, it shows the benefit of using active learning over the pooling approach.
- experiments shows that increased diversity also comes with lower performance


**Weaknesses:**

- the added value of the proposed dataset over the previous one of Founta et al. [31] is not clear. The ratio of tweets with no known hate words is admittedly higher, but in the absolute, since the data set of Founta et al. is much larger, it still contains more of such « diverse » tweets (about 478 tweets vs. 98 in the proposed dataset according to the numbers provided in Table 2 of the paper. The dataset of Founta et al. could thus be re-sampled to reach a better prevalence and relative coverage.
- despite the fact that the authors discuss the limitation of pooling vs. active learning, the proposed dataset IS built using a pooling strategy. It can therefore theoretically not really contain more diversity than the set of datasets on which the models of the pool have been trained
- There is insufficient documentation about several aspects of the dataset (see documentation section this review)


**Additional Feedback:**

About subjectivity vs. ambiguity: it is true that subjectivity may partly explain the observed uncertainty but I think the annotator’s disagreement may also be due to some forms of aleatoric uncertainty, i.e. an intrinsic uncertainty that leads to a non null p(y|x) for a given tweet x.  For instance, a tweet might be intrinsically ambiguous because of some missing information about the context, the author, etc. and the conditional probability might be equal to e.g. p(y|x)=0.6. In such cases, some annotators will say y=1 while others will say y=0 not because they bring a subjective judgement but rather because they have to chose randomly between 1 and 0 . It is very important to keep such ambiguous tweets in the training set because the ambiguous region around the theoretical decision frontier is the most difficult to learn. With this in mind, the proposed approach to select samples (based on pooling or  most positives » samples) might be problematic. Indeed, they are likely to not select such ambiguous tweets and focus on the less ambiguous ones. Classifiers trained on this type of data will then tend to make « black-and-white fallacies » when they encountered ambiguous tweets.

Minors:
«  identifies langauge » —>«  identifies language »

**Clarity:**

the paper is well written


**Documentation:**

There is insufficient documentation about several aspects:
- intended uses / responsible use: the authors should discuss in more details the downstream applications of the models trained on such datasets. Why do we need automated classifiers of tweets containing hate speech ? to censor them ? to signal them ? because they violate the law of some countries? isn't user reporting enough?
- license: there is no mention of any licensing plan (whereas I guess there are some authorship concerns)
- reproducibility: the provided GitHub only contains the dataset but not the models and codes used for the benchmark


**Ethics:**

No strong ethical concerns

**Relation To Prior Work:**

it is sufficiently discussed how this work differs from previous contributions

**Summary And Contributions:**

The paper introduces a new dataset (of about 5K tweets) aimed at training and evaluating hate speech classifiers. The tweets of the dataset were selected via a « pooling » strategy, i.e. by merging the predictions of several models trained on previously existing hate speech datasets. It was then annotated manually based on Amazon Mechanical Turk. Additionally, the paper evaluates a posteriori in what measure an active learning strategy would have been more efficient than this pooling approach.

---

> ### Author Response · Authors · 2021-09-28
> **Reply to Reviewer 1**
>
> We thank the reviewer for their time and helpful feedback. We were glad to see that the reviewer appreciated our point about the bias of existing hate speech datasets that only include tweets matching known hate words, our methods for how to (efficiently) create a more diverse dataset, and our experiments showing that hate speech models perform far worse when known hate words are absent.
>
> **“Why do we need automated classifiers of tweets containing hate speech ? to censor them ? to signal them ? because they violate the law of some countries? isn't user reporting enough?”** Please see a new first paragraph in the Introduction, providing additional background and citing key surveys.
>
> **“the proposed dataset IS built using a pooling strategy. It can therefore theoretically not really contain more diversity than the set of datasets on which the models of the pool have been trained.”** Only including tweets matching a predefined list of known hate words constitutes an inflexible, deterministic rule preventing diversity. In contrast, we start from a uniform random sample of tweets and then probabilistically select tweets to include, allowing inclusion of  tweets lacking known hate words. Even though the datasets used to train our pooling prediction models may lack diverse tweets, the trained models still learn correlations between hate labels and all vocabulary in the dataset. This enables pooling to identify some hateful tweets (for inclusion in our dataset) that lack known hate words, though this prediction task is clearly challenging (as the bottom left of Table 3 indicates). Finally, the purpose of pooling in IR is that diverse models better identify potentially relevant content for human annotation. In the translational use of pooling we present for hate speech, training a cross product of different models across different prior datasets similarly promotes diversity and “wisdom of crowds” in identifying potentially hateful content for inclusion and annotation. We have added this discussion to the Conclusion.
>
> **“...since the data set of Founta et al. is much larger, it still contains more of such diverse tweets...”**. We framed the two extremes across prior methods for selecting tweets to annotate. “At one extreme, random sampling ensures representative coverage but is highly inefficient… At the other extreme, one can annotate only those tweets matching a pre-defined vocabulary of “hate words”... However, it sacrifices representative coverage for cost-savings…”  Our first (translational) contribution is showing how pooling and AL from IR can be used to find diverse forms of hate speech (missed by hate word lists) in a far more efficient manner than random sampling. This methodological contribution is then coupled with a resource contribution: the dataset produced by our method. At large enough scale, even inefficient random sampling will find diverse forms of hate speech, and indeed Founta et al. show this (albeit at high cost). However, our method can be used far more efficiently to create diverse hate speech datasets for other social media platforms, languages, or definitions of hate.
>
> **“There is insufficient documentation about several aspects of the dataset...”** We have added a BSD-3 license. Our experiments use models and datasets from prior work, which we now provide explicit links to, along with a  revised, more accessible README for reproducing results.
>
> **“It is very important to keep ...ambiguous tweets in the training set because the ambiguous region around the theoretical decision frontier is the most difficult to learn. ...the proposed approach to select ...most positive samples might be problematic… they are likely to not select such ambiguous tweets and focus on the less ambiguous ones.”**  In a canonical AL scenario with balanced data, uncertainty sampling selects ambiguous examples for the reason argued here.  However, with highly skewed data (e.g., highly rare relevance in IR or hate speech in social media), prior work in AL for IR has shown empirically the benefit of prioritizing selection of examples predicted to come from the rare class. With highly imbalanced data, it’s important to expose the learner to as many examples of the rare class as possible, and since the model will often be wrong, this will provide ample exposure to the majority class [19]. To clarify this last point, AL prediction of the rare class will often be wrong, and these mistakes (and near-mistakes) will result in sampling many ambiguous examples for both classes. We have clarified this in our Section 2.2. Please also see our discussion of stratified sampling “to further promote diversity” in Section 2.1.

---

### Decision · Program_Chairs · 2021-10-10

**Decision:**

Accept

**Comment:**

This paper presents a benchmark dataset for hate speech detection, using an IR-based approach. Reviewers found the contribution compelling, though questions several aspects of novelty with respect to previous datasets, and methodology.